# Microstructure Observation and Nanoindentation Size Effect Characterization for Micron-/Nano-Grain TBCs

**Haiyan Liu [1], Yueguang Wei [1,*], Lihong Liang [2,*], Yingbiao Wang [3], Jingru Song [3], Hao Long [1] and Yanwei Liu [1]**

1   Department of Mechanics and Engineering Science, College of Engineering, Peking University, Beijing 100871, China; 1706387228@pku.edu.cn (H.L.); mtlonghao@pku.edu.cn (H.L.); 1901111600@pku.edu.cn (Y.L.)

2   College of Mechanical and Electrical Engineering, Beijing University of Chemical Technology, Beijing 100029, China

3   LNM, Institute of Mechanics, Chinese Academy of Sciences, Beijing 100190, China; wangyingbiao123@163.com (Y.W.); songjingru@lnm.imech.ac.cn (J.S.)

*   Correspondence: weiyg@pku.edu.cn (Y.W.); lianglh@mail.buct.edu.cn (L.L.); Tel.: +86-10-6275-7389 (Y.W.)

**Abstract:** Microstructure observation and mechanical properties characterization for micron-/nano-grain thermal barrier coatings were investigated in this article. Scanning electron microscope images demonstrated that both micron-grain coating and nano-grain coating had micrometer-sized columnar grain structures; while the nano-grain coating had the initial nanostructures of the agglomerated powders reserved by the unmelted particles. The mechanical properties (hardness and modulus) of micron-/nano-grain coatings were characterized by using nanoindentation tests. The measurements indicated that the nano-grain coating possessed larger hardness and modulus than the micron-grain coating; which was related to the microstructure of coatings. Nanoindentation tests showed that the measured hardness increased strongly with the indent depth decreasing; which was frequently referred to as the size effect. The nanoindentation size effect of hardness for micron-/nano-grain coatings was effectively described by using the trans-scale mechanics theory. The modeling predictions were consistent with experimental measurements; keeping a reasonable selection of the material parameters.

**Keywords:** micron-/nano-grain coatings; microstructure; nanoindentation size effect; trans-scale mechanics theory

## 1. Introduction

Thermal barrier coatings (TBCs) with low thermal conductivity provide excellent thermal protection and have been widely used to protect hot-components in aero-engines from high-temperature environment [1–3]. The thermal and mechanical properties of TBCs have a significant effect on the lifetime of TBCs. Therefore, the measurements for thermal and mechanical properties are essential. In addition, the thermal and mechanical properties of materials are closely related to their microstructures, so it is worth investigating microstructures of TBCs.

Studies have shown that improvement in physical, thermal, and mechanical properties of new materials can be attributed to improvement of microstructure when compared with conventional materials [4]. For plasma sprayed coatings, particle size of the feedstock powders [5–8] and spray parameters [9–11] have a great effect on the microstructure (such as porosity, micro-crack content, and grain size) of the coatings. Liang et al. reported that the feedstock powders used for depositing

nanostructured coating (particle size of agglomerated powders ranged in 30–50 μm, the size of nanoparticles inside the agglomerated powders varied from 50 to 80 nm) was smaller than the powders used for depositing conventional coating (particle size ranged in 60–100 μm), which contributed to the improvement of microstructure (such as reduction of grain size and contents of pores and micro-cracks) of the nanostructured coating [6]. Bai et al. investigated the effect of spray parameters on the particles temperature and velocity which determined the melting state of the particles, and found that the spray distance had the greatest impact [9]. Their results showed that the content of unmelted nanoparticles decreased from 12% to 6% when the temperature and velocity of in-flight particles increased by 10% and 14%, respectively.

Recently, a great deal of research with respect to the effect of microstructure on thermal properties such as thermal shock (oxidation) resistances [6–8,12,13] and thermal insulation [9,14], and mechanical properties such as hardness and modulus [5,14–22] have been done. Wang et al. found that the nanostructured zirconia coating possessed a better thermal shock resistance than that of the conventional zirconia coating [7]. The enhanced toughness and decreased porosity and micro-cracks in the nanostructured coating contribute to improvement of thermal shock resistance, which is consistent with what Liang reported [6]. Fan et al. discovered the small grain size had a positive influence on thermal fatigue life, and the columnar crystal microstructure had a great effect on thermal shock resistance of TBCs [13]. Wu et al. found that the reduction of porosity resulted in increase of thermal diffusivity, suggesting a significant degradation in the thermal barrier effect [14]. Some researchers have studied the effect of microstructure on hardness and modulus at high temperature [14–18]. They all discovered that the hardness and modulus of TBCs increased with the decrease of porosity caused by sintering of coating at high temperature. Nath et al. evaluated the resistance to plastic deformation of TBC at room temperature and with thermal exposure [15]. Baiamonte et al. indicated that nanostructured coating had larger hardness and modulus than the conventional coating by dynamic indentation tests at room and high temperatures [18]. The effect of microstructure on hardness and modulus at room temperature have also been investigated [19–22]. Jang et al. discovered that the hardness and Young's moduli at the side regions of TBCs decreased with increasing the porosity [19], which is consistent with reference [20]. Lima et al. studied the effect of roughness on the microhardness and elastic modulus of nanostructured zirconia coatings, their results showed that the microhardness and elastic modulus increased with decreasing roughness [21]. Zeng et al. found that the nanocrystalline zirconia coating had a larger microhardness than that of conventional coating, and they used the Hall–Petch model assuming that a dislocation network density slipped through the grain boundaries to explain the phenomenon [22].

In order to study the mechanical properties (hardness and modulus) of TBCs, several tests have been presented and widely used, such as beam bending tests [18,23–25] and indentation tests [5,14–23,26,27]. Thompson et al. indicated that the modulus given by indentation test was much higher than the value based on the beam bending technique [23], which is consistent with Baiamonte reported [18], because indentation test gave an indication of local modulus, and beam bending test gave a global value which contained the effects of micro-crack and porosity. All modulus values measured by beam bending methods were relatively small for TBCs [24,25].

The measured mechanical properties based on beam bending tests are average values on macroscopic scales. However, the measured mechanical properties based on the nanoindentation tests are local values on microscopic scales. The above two types of mechanical properties obtained by the above two scale tests are extremely different. Many studies have shown that an apparent indentation size effect (ISE) of hardness for different kinds of materials was observed, such as brittle materials [26,27], metal materials [28,29], and biomaterials [30]. In order to describe the ISE, a series of different theoretical methods were presented. Li et al. proposed an energy balance analysis based on elastic/plastic indentation model to analyze the apparent ISE of hardness [26]. Zotov et al. used a new empirical equation based on the concept of elastic recovery to explain the phenomenon of ISE [27]. Wei et al. presented the strain gradient plasticity theory based on the thermodynamics frame

to describe the ISE and cross-scale fracture [28,29], and they predicted the micrometer-sized parameter of strain gradient plasticity theory by the comparison of theoretical predictions with experimental measurements. Song et al. [30] used the trans-scale mechanics theory presented by Wu et al. [31] to model the nanoindentation size effect for limnetic shell, and they considered both the strain gradient effect and surface/interface effects.

In this study, the microstructures of micron-/nano-grain coatings were observed by using scanning electron microscopy (SEM) and atomic force microscopy (AFM); the porosity and grain size of micron-/nano-grain coatings were calculated by quantitative image analysis using Image-Pro Plus software. The mechanical properties (hardness and modulus) of micron-/nano-grain coatings were measured by using nanoindentation tests. The effect of microstructure on mechanical properties was analyzed by comparing the microstructure of micron-grain coating with that of nano-grain coating. The trans-scale mechanics theory presented by Wu et al. [31] was adopted to characterize the nanoindentation size effect for the micron-/nano-grain coatings.

## 2. Experimental Procedure

### 2.1. Materials and Plasma Spray

Ni-based superalloy plates (GH3128) with a dimension of 250 mm × 50 mm × 1.2 mm were used as substrates. A commercial CoNiCrAlY powder (AMDRY 995, Sulzer Metco, Pfäffikon, Switzerland) with a grain size of 5–37 μm was used for depositing the bond coat. Two kinds of spray-dried and sintered $ZrO_2$-8 wt % $Y_2O_3$ (8YSZ) powders were used as the feedstock for spraying on the bond coat to form the top coat with 300 μm and 500 μm in thickness. The particle size of the first feedstock ranged from 60 to 100 μm, the small particles inside the feedstock were submicron particles with a grain size of 320–650 nm. In the present research, the micron-grain coating was produced from the first feedstock. The particle size of the second feedstock varied from 30 to 50 μm, the small particles inside the feedstock were nanoparticles with a grain size of 10–50 nm. The nano-grain coating was fabricated from the second feedstock. The spraying process was carried out by atmospheric plasma spraying (APS) system (A-2000, Sulzer-Metco F4 gun, Pfäffikon, Switzerland). The coating samples were fabricated by the Shanghai Institute of Ceramics, Chinese Academy of Sciences, Shanghai, China. The parameters for plasma spraying were given in Table 1.

**Table 1.** Plasma spray parameters.

| Parameters | NiCrCoAlY Bond Coat | 8YSZ Coating |
|---|---|---|
| Primary Ar (slpm) | 57 | 30 |
| Second $H_2$ (slpm) | 8 | 12 |
| Spray distance (mm) | 120 | 120 |
| Gun current (A) | 620 | 620 |
| Power (kW) | 39 | 42 |
| Carrier gas (slpm) | 3.5 | 3.5 |
| Powder feed rate (g/min) | 40 | 40 |

### 2.2. Microstructure Observation Tests

The samples for microstructure observation on surface were circle plates with polished coating surface, diameters of 14 mm and top coat thicknesses of 300 μm. The samples for microstructure observation on cross section were cuboids with polished (fractured) cross section, lengths of 20 mm, widths of 3 mm, and top coat thicknesses of 500 μm (300 μm). The surface and cross-sectional microstructure for micron-/nano-grain coatings were observed by using SEM (JSM-7500F, JEOL, Tokyo, Japan), and the micrographs of unmelted nanoparticles were observed by using AFM (MultiMode 8, Bruker, Billerica, MA, USA).

### 2.3. Nanoindentation Tests

The samples for nanoindentation experiments were circle plates with polished coating surface and diameters of 14 mm. Thicknesses of top coat, bond coat, and substrate were 300 μm, 15 μm, and 1.2 mm, respectively. Hardness and elastic modulus were measured on the coating surface of micron-/nano-grain coatings based on the nanoindentation tests. The tests were carried out using a Nano Indenter G200 (Agilent, Santa Clara, CA, USA) with a diamond Berkovich tip (tip curvature radius is about 20 nm). The force and displacement resolutions of G200 are about 1 nN and 0.0002 nm, respectively. The tests were performed under displacement control with a maximum indent depth of 1 μm. Hardness (*H*) and elastic modulus (*E*) were obtained based on the Oliver–Pharr method [32], the corresponding expression was

$$H = \frac{P}{A}, \ \frac{1}{E_r} = \frac{1 - v^2}{E} + \frac{1 - v_i^2}{E_i},$$

(1)

where *P* was the indent load, *A* was the contact area, $E$ ($E_i$) and $v$ ($v_i$) were the elastic modulus and Poisson's ratio for the sample (indenter), respectively. The material parameters of the diamond indenter were taken as $E_i = 1141$ GPa and $v_i = 0.07$. The reduced elastic modulus ($E_r$) was obtained using a relation including contact stiffness and contact area.

## 3. Experimental Results and Discussion

### 3.1. Microstructure Characteristics

Figure 1 shows the surface images of as-sprayed micron-/nano-grain coatings. Microscopically, the coating was non-uniform and non-dense, and had a large quantity of pores and micro-cracks. As shown in Figure 1, the nano-grain coating had a denser microstructure and fewer irregular pores and micro-cracks than the micron-grain coating. The surface microstructure characteristics are related to the formation mechanism of coating. In the process of forming the coating, the ceramic powders were first melted or partially melted under the plasma flame and rushed toward the surface of bond coat at high speed. The powders deformed and rapidly condensed and shrank to adhere on the bond coat in flat state. Because of insufficient deformation of some ceramic particles, pores generated among the particles, resulting in non-uniform and non-dense structure. The higher porosity of micron-grain coating may be attributed to the larger particle size of feedstock. The occurrence of micro-cracks on the coating surface was mainly due to the large tensile stress generated when the ceramic droplets condensed and shrank.

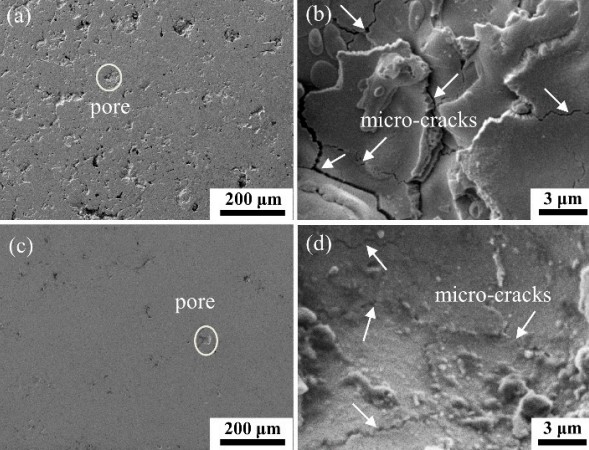

**Figure 1.** Microstructure of coatings: (**a**) polished surface of micron-grain coating; (**b**) a detailed view of the micro-cracks of micron-grain coating in circular region of (**a**); (**c**) polished surface of nano-grain coating; (**d**) a detailed view of the micro-cracks of nano-grain coating in circular region of (**c**). The magnifications of images (**a**–**d**) are 100, 5000, 100, 5000×, respectively.

Figure 2 shows the polished cross-section of representative specimens for micron-/nano-grain coatings. From the figure, it can be seen that the TBC system was clearly a multilayer structure, with many pores distributed randomly in the top coat. In this article, Image-Pro Plus software (Media Cybernetics, Silver Springs, MD) was utilized to calculate porosity by quantitative image analysis. The final statistical result of porosity was the average value based on 10 cross-sectional images for an individual sample [8,9]. The statistical results showed that the porosities of micron-/nano-grain coatings were approximately 6.5% ± 0.2% and 5.1% ± 0.3%, respectively.

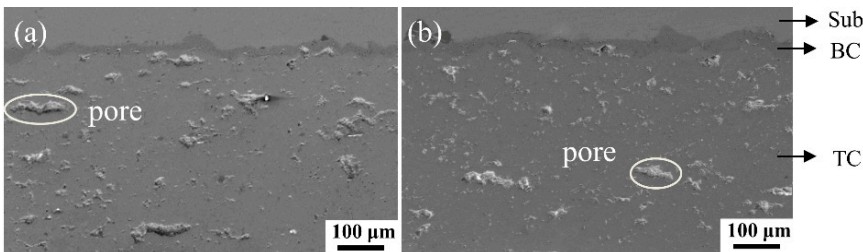

**Figure 2.** Images of polished cross-section of specimens: (**a**) for micron-grain coating; (**b**) for nano-grain coating. Sub, substrate; BC, bond coat; TC, top coat. The magnification of images (**a**) and (**b**) is 350×.

Figure 3 shows the images of fractured cross-section of micron-/nano-grain coatings. As seen from Figure 3a, the micron-grain coating consisted of layered structures (splats) with many inter-splat cracks, the micrometer-sized columnar crystals within the splats (see Figure 3b) were observed, and long cracks among the columnar crystals were visible. Rapid nucleation occurred at the cooler surface of the flattened droplet at large undercooling and the crystals grew rapidly opposite to the heat flow, leading to the formation of a columnar grain structure [8]. The columnar microstructure showed a strong anisotropy due to different heat flow directions, as reported by [19]. Figure 3c,d show the images of fractured cross-section of nano-grain coating. From the figures, columnar grain structures were also observed, inter-splat cracks and long cracks were also visible (see Figure 3c). The average inter-splat spacing of the nano-grain coating was smaller than that of the micron-grain coating. Fewer and smaller inter-splat cracks and long cracks existed in nano-grain coating, that is to say the nano-grain coating had a finer columnar grain structure. However, the nano-grain coating had a significantly different feature compared with the micron-grain coating, because some unmelted particles were loosely distributed in the recrystallization zone (splats) as shown in Figure 4, which is consistent with literatures [5,7–9]. These unmelted particles reserved the initial nanostructure of the agglomerated powder, as later confirmed by AFM (see Figure 5e). The complex structure of nano-grain coating was related to the formation mechanism of the coating. Individual submicron-sized and nanosized powders cannot be carried in a moving gas stream and deposited on a substrate because of their low mass and flowability. To overcome the above problem, the slurries prepared from the submicron-sized and nanosized powders was then spray dried to form micrometer-sized agglomerated powders. The two agglomerated powders were used as the feedstock for depositing the top coat. All sprayed feedstock were accelerated by the plasma flame, then rushed towards the surface of the bond coat at high speed and deposited. The melted particles re-nucleated and grew in the liquid state, forming the columnar grain structures (see Figure 3d); the unmelted particles reserved the initial nanostructure of the agglomerated powders (see Figure 4). The columnar grain structure acted as a binder and maintained the integrity of the coating.

During the plasma spraying, the particle temperature and velocity determined the melting state of particles. The plasma spray parameters affected the temperature and velocity of in-flight particles, such as powder feed rate, spray distance, spray power [9]. In the present research, we used the same spray parameters to deposit the two kinds of top coats, only considered the influence of particle size of feedstock powders on the microstructure of as-sprayed coatings. The effect of particle size of feedstock

powders on the pore, micro-crack, and columnar grain structure was illustrated in Figures 1–3, and on grain size of as-sprayed coatings was demonstrated in Figure 5, as discussed in the following text.

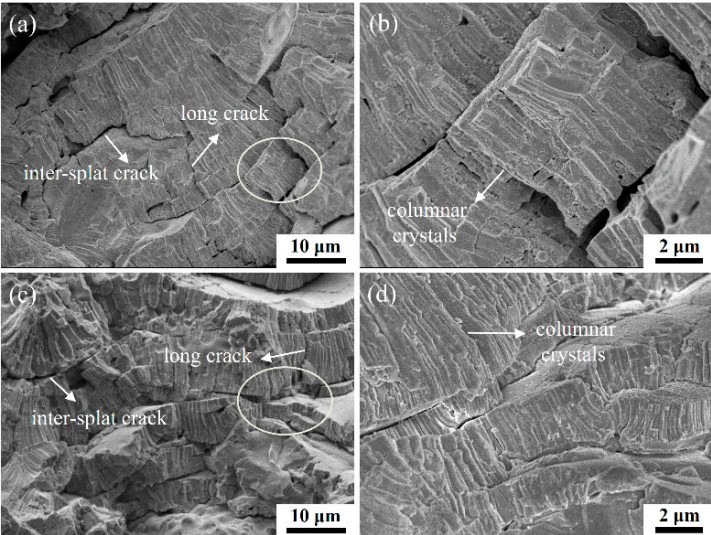

**Figure 3.** Microstructure of coatings: (**a**) fractured cross-section of micron-grain coating; (**b**) a detailed view of the columnar crystals of micron-grain coating in circular region of (**a**); (**c**) fractured cross-section of nano-grain coating; (**d**) a detailed view of the columnar crystals of nano-grain coating in circular region of (**c**). The magnifications of images (**a**–**d**) are 2000, 8000, 2000, 8000×, respectively.

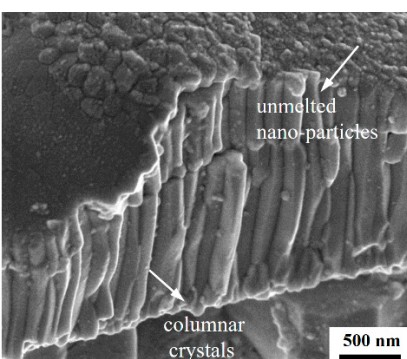

**Figure 4.** SEM image of columnar crystals and unmelted nano-particles in the nano-grain coating. The magnification of the image is 30,000×.

Figure 5 shows the representative images of grains in the micron-/nano-coatings, and distributions of grain size. The grain sizes were calculated by quantitative image analysis using Image-Pro Plus software. The final statistical result of grain sizes was the average value based on five images for an individual sample. The SEM image illustrated in Figure 5a indicates that micron-grain coating was composed of grains with different size (size ranged from 393 to 2507 nm), some micro-pores and micro-cracks were found at the intersection of multiple grains. The statistical result shown in Figure 5b demonstrates that the grain sizes were distributed mainly in 500–1500 nm, and the average value of grain size of micron-grain coating was approximately 1123 ± 310 nm. Figure 5c gives the SEM image of nano-grain coating surface after thermal etching, the image reveals that the coating was composed of small grains (size ranged in 75–96 nm) and large grains (size varied from 101 to 984 nm). Figure 5d reveals that the grain sizes were distributed mainly in 75–400 nm, and the average value of grain size of nano-grain coating in the recrystallization zone was approximately 242 ± 87 nm. Figure 5e shows the AFM image of nanograins in the nano-grain coating. The sizes of unmelted nanoparticles ranged in 10–50 nm, revealing these unmelted nanoparticles maintained the initial nanostructure of

agglomerated powder. Figure 5f indicates that the grain sizes were distributed mainly in 10–30 nm, and the average value of grain size of nanoparticles was approximately 20 ± 6 nm. It was clear from Figure 5a–d that the grain size of nano-grain coating in the recrystallization zone was smaller than that of the micron-grain coating. The reduction of particles size of feedstock powders contributed to the reduction of grain size of the coating, leading to the improvement of microstructure of the coating.

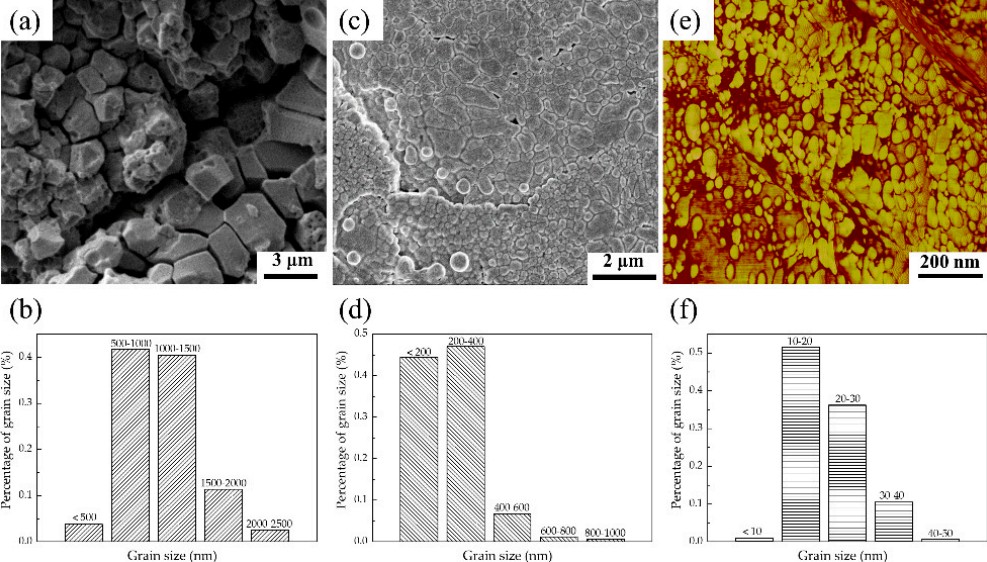

**Figure 5.** (**a**) SEM image of grains in the micron-grain coating; (**b**) statistical results on grain size of the micron-grain coating; (**c**) SEM image of grains in the recrystallization zone of nano-grain coating; (**d**) statistical results on grain size of nano-grain coating in the recrystallization zone; (**e**) AFM image of nanograins in the nano-grain coating; (**f**) statistical results on grain size of unmelted nanoparticles in the nano-grain coating. Numbers on the *y*-axis indicate the size of grains. The magnifications of images (**a**) and (**c**) are 5000 and 10,000×, respectively.

### 3.2. Hardness and Modulus Measurements

In the present research, the nanoindentation mechanical properties (hardness and elastic modulus) of micron-/nano-grain coating were measured and compared. It is worth noting that in the nanoindentation test, the influence of indenter tip curvature on the hardness measured is inevitable and a challenge question in that area. However, it is meaningful to distinguish mechanical properties between nano-grain coating and micron-grain coating by using the technique in averaging meaning. Figure 6 demonstrates the hardness-depth curves of the two kinds of TBCs corresponding to 10 indent points on the surface of top coat. Curves shown in Figure 6a,b correspond to the cases of micron- and nano-grain coating, respectively. When the indent depth was very shallow, the hardness-depth curves had a dramatic variation resulted from many factors, such as the unsteady contact between the indenter and the specimen surface at the beginning of the test, surface morphology of the indenter and specimens, etc. However, as the indent depth increased, the hardness-depth curves tended to keep stable rapidly. From Figure 6, generally the measured hardness-depth curves varied with indent depth non-monotonically because of radius of curvature existed in the tip of the indenter. When the indent depth was larger than about 100 nm, the influence of indenter on variation of the curve can be ignored, and the hardness increased with the decrease of indent depth, this phenomenon was called "nanoindentation size effect". The measured hardness values of nano-grain coating were larger than those of micron-grain coating at the same indent depth by comparison of curves in Figure 6a with those in Figure 6b. In order to compare the mechanical properties of the two coatings more clearly, we further derived the average value of hardness and modulus based on the curves and displayed the

results in Figure 7. Figure 7 shows that both the measured average values of hardness and modulus of nano-grain coating were larger than those of micron-grain coating.

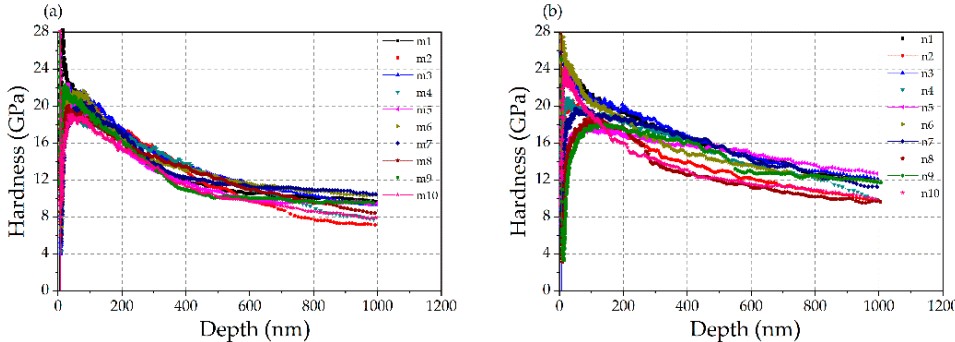

**Figure 6.** Hardness–depth curves of the two kinds of TBCs corresponding to 10 indent points on the surface of top coat, m and n denote micron-grain coating and nano-grain coating, respectively, (**a**) for micron-grain coating, (**b**) for nano-grain coating.

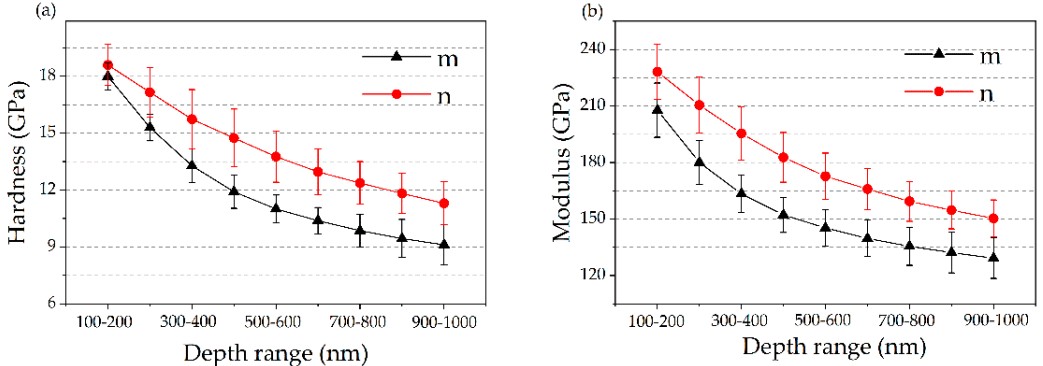

**Figure 7.** Average hardness and modulus of two kinds of TBCs in different depth ranges based on 10 indent points, m and n denote micron-grain coating and nano-grain coating, respectively, (**a**) for hardness and (**b**) for modulus.

As we all know, the mechanical properties of materials are closely related to their microstructures. In the present research, the feedstock powders used for depositing nano-grain (OSe) coating was smaller than those for depositing micron-grain coating, which contributed to the improvement of microstructure (such as reduction of contents of pores and micro-cracks and reduction of grain size) of the nano-grain coating, as Liang reported [6]. Furthermore, microstructure observation shows that the nano-grain coating was denser and had finer columnar grain structure and fewer pores and micro-cracks than the micron-grain coating (see Figures 1–3), so hardness and modulus of nano-grain coating were larger than those of the micron-grain coating, as reported by references [5,18]. In addition, the nanostructures and reduction of grain size of nano-grain coating (see Figure 5) contributed to the increase of hardness of the coating according to the Hall–Petch equation.

## 4. Modeling of Nanoindentation Size Effect for Micron-/Nano-Grain Coating

### 4.1. Theoretical Model Based on the Trans-Scale Mechanics Theory

Considering that the coatings has a certain inelastic deformation when the nanoindentation depth is at the nanoscale, and assuming that the coating is continuous and homogeneous, without considering the effects of pores and micro-cracks, we use the trans-scale mechanics theory [31] to describe the nanoindentation size effect, including the strain gradient and surface/interface effects, and the strain gradient effect is described by strain gradient plasticity theory.

For micron-grain coating, only the strain gradient effect is considered assuming that the surface/interface effects can be neglected in this case. For nano-grain coating, both strain gradient and surface/interface effects are considered.

The trans-scale mechanics theory only considering strain gradient effect was studied by Wei et al. [28], the expression of solution can be written as

$$\frac{H_{SGP}}{H_0} = g_0(\frac{h}{l}, \frac{E}{H_0}, \nu, N, \beta) \tag{2}$$

where $H_0$ is the reference hardness value corresponding to deep indentation hardness, $l$ is material length scale, $h$ is indent depth, $N$ is strain hardening exponent of material, $\beta$ describes the angle of the circular conic indenter through area equivalency with true pyramid indenter.

The trans-scale mechanics theory considering strain gradient and surface/interface effects was investigated by Song et al. [30], the nanoindentation hardness $H$ with respect to indent depth $h$ for the case of circular conic indenter and TBCs can be expressed as

$$\frac{H}{H_0} = f(\frac{h}{l}, \frac{E}{H_0}, \nu, N, \beta, \frac{\gamma}{H_0 h}, \frac{\Gamma}{H_0 d_0}) \tag{3}$$

where $\gamma$ is the surface energy density, $\Gamma$ is the interface energy density, $d_0$ is the representative size of the nanoparticle. The solution for Equation (3) can be obtained by quantity level analysis. Since the values of the dimensionless quantities both $\gamma/H_0 h$ and $\Gamma/H_0 d_0$ are much smaller than unity within an effective indent depth range, the dimensionless hardness can be approximately written as follows through small parameter expansion for the case of circular conic indenter and equal-size cubic grain [30]

$$\frac{H}{H_0} \approx g_0(\frac{h}{l}, \frac{E}{H_0}, \nu, N, \beta) + \frac{2}{\cos \beta} \frac{\gamma_s}{H_0 h} + \frac{6\sqrt{3}\Gamma}{H_0 d_0 \tan \beta}(1 - \frac{d_0 \tan \beta}{2\sqrt{3}h}) \tag{4}$$

where the first term of right-hand side is the solution only considering the strain gradient effect (see Equation (2)), and the other terms are the contributions resulted from the surface and interface effects.

*4.2. Interpretation to Experimental Hardness Size Effect Based on the Theoretical Model*

Hardness-depth curves shown in Figure 6 can be also simulated by using the trans-scale mechanics theory through considering a curvature radius existing at indenter tip. $H_0$ for deep indent is taken as 2.8 GPa for the present case. For the circular conic indenter, $\beta$ is taken as 30° [28]. For the nano-grain coating used in experiments, the nanoparticle size $d_0$ can be measured based on the AFM figure and is about 20 nm. Coating can be considered as a medium-level hardening material when the nanoindentation depth is at the nanoscale, so we take $N = 0.3$.

Figure 8 shows the theoretical simulations to experimental results for two kinds of TBCs. In Figure 8a, two experimental curves denoted by symbols corresponding to the upper bound and lower bound curves in Figure 6a for micron-grain coating, respectively, theoretical results denoted by black solid lines are based on Equation (2). In Figure 8b, two experimental curves denoted by symbols corresponding to the upper bound and lower bound curves in Figure 6b for nano-grain coating, respectively, theoretical results denoted by black solid lines are based on Equation (4). For the micron-grain coating, we only consider the strain gradient effect, one can obtain the value of material length scale by comparison of theoretical results with experimental ones ($l = 4.81$–$7.51$ μm), and the values are within reasonable ranges [33]. For the nano-grain coating, substituting the material and geometric parameters into Equation (4), and letting the values of surface energy density and interface energy density the same in this paper, one can obtain the values of material length scale and surface/interface energy density by comparison of theoretical results with experimental ones ($l = 1.73$–$7.26$ μm, $\gamma = \Gamma = 2.88$–$7.12$ J/m$^2$), and the values are within reasonable ranges [33]. Therefore, the trans-scale theory can effectively model the nanoindentation size effect for the two kinds of TBCs.

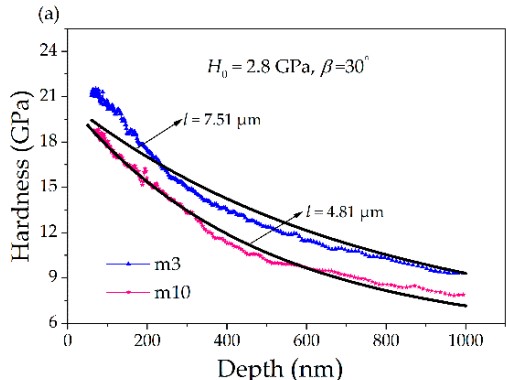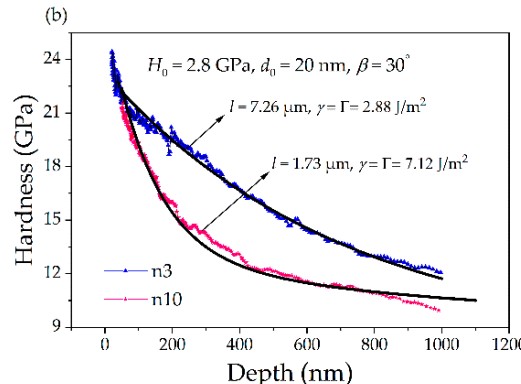

**Figure 8.** Theoretical simulations to experimental results for two kinds of TBCs. (**a**) For micron-grain coating, two experimental curves denoted by symbols corresponding to the upper bound and lower bound curves in Figure 6a, respectively, theoretical results denoted by black solid lines are based on Equation (2). (**b**) For nano-grain coating, two experimental curves denoted by symbols corresponding to the upper bound and lower bound curves in Figure 6b, respectively, theoretical results denoted by black solid lines are based on Equation (4).

## 5. Conclusions

For two kinds of TBCs, experimental microstructure observation, hardness, and modulus measurement by using the nanoindentation tests were performed, and nanoindentation size effect was described by using the trans-scale mechanics model. The main conclusions can be summarized as follows:

- Scanning electron microscope images reveal that the micron-grain coating had a columnar grain structure with an average grain size of about 1123 nm; the nano-grain coating also had a columnar grain structure formed by recrystallization of the melted particles with an average grain size of about 242 nm, while it possessed the initial nanostructure of the agglomerated powders reserved by the unmelted particles with an average grain size of about 20 nm.
- The hardness and modulus of two kinds of TBCs were measured by using nanoindentation tests. The measurement relation between hardness and indent depth showed a strong size effect. The measured results indicated that the nano-grain coating had larger hardness and modulus than the micron-grain coating, the improved properties of nano-grain coating were associated with retained nanostructure and reduction of porosity, micro-crack, and grain size of the coating.
- The nanoindentation size effect of hardness for the two kinds of TBCs was effectively described by utilizing the trans-scale mechanics theory, the values of material parameters were obtained by comparison of theoretical predictions with experimental measurements and they were within reasonable range.

**Author Contributions:** Data curation, Y.W. (Yingbiao Wang) and Y.L.; Methodology, J.S. and H.L. (Hao Long); Writing—original draft, H.L. (Haiyan Liu); Writing—review and editing, Y.W. (Yueguang Wei) and L.L. All authors have read and agreed to the published version of the manuscript.

**Funding:** This research was funded by the National Natural Science Foundation of China, Nos.: 11890681, 11672301, 11511202, 11672296, 91860102, 11432014.

**Conflicts of Interest:** The authors declare no conflict of interest.

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
