# Peer review of "Microstructure Observation and Nanoindentation Size Effect Characterization for Micron-/Nano-Grain TBCs"

_coatings, doi:10.3390/coatings10040345_

Round 1

Reviewer 1 Report

Dear Authors,

In the following, I summarize my comments to your manuscript about microstructural observations and nanoindentation in micro- and nano-grained TBC coating:

Line 81 to 83: Rewrite the sentence.

Line 93 to 98:

Several information is missing, which is also important to classify the presented results.

  • Include details of the used NiCrAlY and micro-/nano sized TBC powders, like manufacturer, particle size of the used powders, especially the agglomerated powders including the size of the micro- and nanoparticles of the starting materials.
  • Include the used thermal spraying process for the coating deposition and used spray parameters. Contrary to your referenced work [27], you only used the air plasma spray method in this work as well as other coating thicknesses.
  • Please confirm that the NiCrAlY coating thickness is only 10 µm.

Line 100:

Coating thickness of 300 µm: Does it refer to either the coating system or only the top coat?

Line 104 to 168:

In these lines, experimental observations are described and discussed. I recommend including these details in the Section 3 Experimental results and discussion.

Line 104 to 116:

Please include, how and where have you measured the porosity of the coatings?

Line 120:

Please rewrite the whole sentence (expression): The formation of columnar grain structure was because …

Line 124 to 126: The sentence would be easier to read, when it comprises of two sentences.

Line 134:

poor fluidity --> Normally, flowability is used when dealing with powders.

Figure 4:

Use the same unit at x-axis for the grain size of the micro- and nano-grained coatings.

Line 184 following and Figure 5:

Structure the section according the one above: First, mention the micro-grain and then the nano-grained coatings.

Some general mistakes:

Sometimes you use micron-grain, sometimes mirco-grain or micro-scale. Please check the spelling.

Often you write power instead of powders. Please check the spelling.

Kind regards

Reviewer 2 Report

The list of detailed remarks is given below:

  1. The bond coat thicnkess at the level 10 microns is in my opinion to small.
  2. What was the particles size of micro- and nano-grain powder? I mean the particle, not grain.
  3. If the particles were such small, please describe the manner of such small particles injection to the plasma jet.
  4. There is no information about the spraying parameters.
  5. What was the porosity measurement method? What was the standard deviation? How many measurements carried out?
  6. How Authors evaluated the porosity only from top-view photographs?
  7. If Authors used optical analysis, please add the magnification of the SEM photos.
  8. Fig. 4 (b) and (d) there is an error on the Y-axis, it should be dozen of %, moreover, how many photos was used to create such histogram?
  9. In line 172 Authors wrote that hardness measurement carried out on coatings surface, whereas in Fig. 5 there are hardness-depth of coatings cross-sections. Please specify it.
  10. Fig. 6 - there are bigger standard deviation bars for nano-grains coatings, than for micro- ones. Could Authors decribe this phenomenon?

Reviewer 3 Report

The authors chose to evidence the effect of microstructural features of thermal barrier coatings on the hardness and Young's modulus. While the conclusions are evidenced by the results, there is little information on how these results were obtained. The level of english is, at times, difficult to understand as well and some parts need to be reformulated. Further comments can be found below: 

General

  • The introduction is too general and should be improved. Think of adding more infromation regarding the material selected, grain sizes and actual values from the researchers that are cited. 
  • The microstructural observations and their description should not be included in section 2. Experimental observation and measurement but rather have a separate section in 3. Results and discussions
  • Different fonts and sizes were used at random places within the text. 
  • Most of the text need to be re-worked on and some parts must be reformulated completely.

Section 2.1 Microstructure observation.

  • There is no information on the grain sizes of the powders used. Having an idea of the initial materials would be helpful to explain the growth of the coatings. I would suggest including SEM micrographs of the powders used before spraying. 
  • There is also no information on the spraying parameters and how they are comparable with respect to the initial powder grain size. In fact, the authors only mention: "During the plasma spraying, many factors affected the temperature and velocity of in-flight agglomerated  powders, such as spray power, spray method and parameters.." They should develop more on this matter and give their deposition parameters in order to correlate observations with the process used and the "formation mechanism of the film". 
  • The references of the SEM and AFM used are missing. 
  • Furthermore, the references of the SEM and AFM are missing. 

Section 3. Experimental results and discussions

  • This section should include the microstructural observations.
  • One of the examples of the sentences and statements that MUST be reformulated: "When the indent depth was larger than about 100 nm, hardness increased with decreasing the indent depth, this change rule showed obvious size effect." 
  • The authors mention that the nano-grain coatings have a higher hardness because of lower pore and crack contents, their finer columnar mictrosture and because of the "nanostructure" retained by the nano-grains. I would like for the authors to develop this more (see above - describe starting powders).
  • Finally, the tip used for the nano-indentation tests is of 20 nm, therefore, one can assume that several grains are affected by the tip in the case of the nano-grain coating. Is there an approximation on the number of grain boundaries that are encountered during one test? How does that number relate to the one of the micro-grain coating?

Section 4.2 Interpretation of experimental hardness size effect based on the theoretical model

  • Page 9, "For the nano-grain coating, substituting the material and geometric parameters into Eq. (3), and letting the values of surface energy density and interface energy density the same in this paper, one can obtain the values of material length scale and surface/ interface energy density by comparing the theoretical and  experimental results.." which surface and interface energy densities were used?
  • Finally, in Figure 7(a) the experimental results and the theoretical ones fit very nicely indicating that the model and the parameters used are descriptive of the material system. However, Figure in 7(b) the fit is somewhat off, particularly for the higher boundary. Would taking into account surface and interface effects change significantly the fit?

Round 2

Reviewer 1 Report

Dear Authors,

The paper was well improved. Some minor spelling and formatting has to be improved before final publication.

Thanks a lot and good luck.